# p38 Molecular Targeting for Next-Generation Multiple Myeloma Therapy

**DOI:** 10.3390/cancers16020256

**Published:** 2024-01-06

**Authors:** Mario Morales-Martínez, Mario I. Vega

**Affiliations:** 1Molecular Signal Pathway in Cancer Laboratory, UIMEO, Oncology Hospital, Siglo XXI National Medical Center, Mexican Institute of Social Security (IMSS), Mexico City 06720, Mexico; 2Department of Medicine, Hematology-Oncology and Clinical Nutrition Division, Greater Los Angeles VA Healthcare Center, UCLA Medical Center, Jonsson Comprehensive Cancer Center, Los Angeles, CA 90095, USA

**Keywords:** multiple myeloma, p38, transcriptional factor, cancer progression, clinical implication, gene expression

## Abstract

**Simple Summary:**

In the last two decades, the use of small molecules as chemical inhibitors of different targets in signaling pathways has been of great interest for therapeutic purposes. In the case of multiple myeloma, it has not been an exception, with researchers focusing in part on the p38 pathway. However, the search continues for more specific p38 inhibitors according to differently expressed p38 isoforms. One of the latest reviews on this topic is from 10 years ago, so in this review, we highlight the research in this context from the last 10 years, showing the advances in the use of more effective inhibitors alone or in combination with other therapeutic agents.

**Abstract:**

Resistance to therapy and disease progression are the main causes of mortality in most cancers. In particular, the development of resistance is an important limitation affecting the efficacy of therapeutic alternatives for cancer, including chemotherapy, radiotherapy, and immunotherapy. Signaling pathways are largely responsible for the mechanisms of resistance to cancer treatment and progression, and multiple myeloma is no exception. p38 mitogen-activated protein kinase (p38) is downstream of several signaling pathways specific to treatment resistance and progression. Therefore, in recent years, developing therapeutic alternatives directed at p38 has been of great interest, in order to reverse chemotherapy resistance and prevent progression. In this review, we discuss recent findings on the role of p38, including recent advances in our understanding of its expression and activity as well as its isoforms, and its possible clinical role based on the mechanisms of resistance and progression in multiple myeloma.

## 1. Introduction

Multiple myeloma (MM) is a hematologic malignancy that remains incurable, as most patients eventually relapse or become refractory to treatment [1]. Even though recent treatments have been improved, resistance to treatment persists in MM [2]. New therapeutic agents have recently been developed for the treatment of resistant or refractory MM, including immunomodulatory agents, proteasome inhibitors, monoclonal antibodies, and therapies directed at molecular targets in different signaling pathways. These therapeutic alternatives have shown relative antitumor activity in resistant and refractory MM, and recent combinations of these therapies have shown clinical effectiveness [3]. Recent studies have focused on therapies directed at molecular targets in refractory or relapsed MM, which are based on signaling pathways specifically activated in tumor cells, as is the case for p38 [3]. Mitogen-activated protein kinases (MAPK) allow cells to respond to a wide variety of stimuli and signals such as DNA damage or inflammatory cytokine activity, and extracellular stimuli such as oxidative and osmotic stress and heat shock [4,5]. p38 has four well-characterized isoforms: p38α, β, γ, and δ [6]. These four isoforms participate in the translation of signals that play a very important role in different cellular processes, such as cell proliferation, differentiation, glucose metabolism, and lipid secretion; senescence; the stress response; apoptosis; autophagy; and cell migration [7,8,9]. Depending on the context, p38 can act as a tumor promoter or tumor suppressor [7,10]. p38 is constitutively activated in MM and has been implicated in osteoclast and osteoblast activity and bone destruction [11]. Recent studies have shown that treatment with bortezomib can induce p38 activation, and this activation may be related to chemoresistance [12,13]. However, other studies show that p38 activation is associated with the induction of apoptosis and autophagy in MM [14]. In this review, we discuss recent advances in our understanding of the regulation and expression of p38, as well as its activity in MM, its possible role as a therapeutic target, and its clinical implications. To construct the introduction of this review, general bibliographies of papers on the molecular aspects of p38 were consulted, as well as review articles from the last 20 years. For references on the medullary part of p38 in MM, articles from the last 10 years were consulted with the keywords “p38 in MM”, “inhibitors of p38 in MM”, and “p38 regulation mechanism” in the PubMed and Google scholar databases in English. A total of 80 publications published in the last 10 years with the words “p38 in MM” were obtained and their content was analyzed, and only five, which did not highlight the importance of p38 in MM, were discarded. A total of 95 publications on p38 inhibitors in MM were found, and 45, which highlighted the importance of p38 and its inhibition in the response of MM, were analyzed and considered.

## 2. p38 Molecular Signaling

The p38 family is a family of specific protein serine/threonine kinases characterized by a dual Thr-Gly-Tyr phosphorylation motif. Four isoforms have been identified, each encoded by four genes with high sequence homology: *MAPK14*, *MAPK11*, *MAPK12*, and *MAPK13*, which code for p38α, p38β, p38γ, and p38δ, respectively [15]. They share 75% homology at the amino acid level between p38α and p38β and between p38γ and p38δ [16]. On the other hand, p38γ shares 62% homology with p38α, whereas p38δ shares 61% homology with p39α [6,17].

p38α is one of the four isoforms that is expressed in all cells and tissues, so it is believed that it plays a very important role in the different cellular signaling cascades triggered by extracellular stimuli such as stress, proinflammatory cytokines, or direct transcription activation. P38α acts as an integration point for multiple biochemical signals and is involved in a wide variety of cellular processes, including proliferation, differentiation, the regulation of transcription, and development. This kinase is activated by various environmental stresses and proinflammatory cytokines. Its activation requires phosphorylation by MAP kinase kinases (MKKs), or autophosphorylation triggered by its interaction with the MAP3K7IP1/TAB1 protein [18,19].

The phosphorylation of p38α can activate a wide range of stimuli, such as transcription factors, protein kinases, cytoplasmic substrates, and nuclear substrates [15]. Unlike p38α, p38β is mainly expressed in the brain [20]. Although p38α and p38β have been described as having shared functions, p38β is less expressed in different cell types. In general, p38α and p38β participate in cellular processes such as cell proliferation and differentiation, glucose and lipid metabolism, secretion, the stress response, apoptosis, autophagy, and cell migration [9]. Under environmental stress, p38β is activated by phosphorylation mediated by MAP2K3/MKK3, MAP2K4/MKK4, and MAP2K6/MKK6. In addition, p38β selectively interacts with histone deacetylase HDAC3 in order to repress ATF2 activity, regulating TNF expression after LPS stimulation [21].

p38γ is mainly expressed in the skeleton, and p38δ in the liver, pancreas, small intestine, and testes [22,23]. Therefore, their expression is more restricted, which is why they are believed to have more specialized functions [5]. There is currently no evidence that the p38γ and p38δ isoforms perform the same functions as p38α; however, in certain cell lines, these isoforms carry out the same functions of tissue regeneration and immune response [16]. Although the downregulation of p38α has been shown to induce the expression of p38γ and p38δ, the individual or combined function of the four p38 isoforms has not been described in detail [24]. p38γ signaling also positively regulates the expansion of transient amplifying myogenic precursor cells during muscle growth and regeneration [21,22,25]. p38δ is another of the four p38s that play an important role in cascades of cellular responses elicited by extracellular stimuli such as proinflammatory cytokines or physical stress, leading to the direct activation of transcription factors [26,27,28]. p38δ is one of the least studied isoforms of p38; its downstream targets include MAPKAPK2, which is phosphorylated and, in turn, phosphorylates subsequent targets. p38δ has been described to have a role in the regulation of protein translation through regulation of the EEF2K protein, as well as in cytoskeletal remodeling through the activation of MAPT and STMN1. 

In a tumor context, the expression of the four isoforms varies and depends on the type of cancer. For example, significant levels of expression of p38α and p38δ have been reported in the primary tumors of most cancers, while p38β and p38γ have significantly decreased expression in the primary tumors of breast cancer, lung adenocarcinoma, and glioblastoma multiforme [9]. Other studies have reported that non-Hodgkin’s lymphoma (NHL) patient samples and cell lines show a significant correlation of p38α expression with malignancy [29].

Studies have shown that p38α induces the activation of pro-inflammatory and apoptosis-inhibiting signals, such as IL-6, IL-8, and IL12b, promoting cell survival and resistance to chemotherapy such as in MM, allowing post-treatment DNA repair [30]. Studies have reported that p38 also plays a role in cell invasion, by inducing epithelial mesenchymal transdifferentiation [31] and inhibiting the matrix metalloproteins MMP-2 and MPP-9, which promotes metastasis and tumor invasion. Additionally, it has been shown that p38 promotes cell migration after the induction of VEGF expression. [32,33].

### 2.1. p38 Structure

*MAPK14* (p38α): This gene is located on the short arm of chromosome 6 (6p21.31), (GRCh38/hg38by Entrez Gene) and consists of 96,433 base pairs in a plus-strand orientation. It encodes a 360-amino-acid protein (MW = 41,293 Da) (NCBI Entrez Gene 1432). In its sequence, it contains a protein kinase domain at residues 24–308 and a TXY phosphorylation motif at positions 180–182 (Figure 1). There are two ATP-binding sites at positions 30–38 and 53. Several post-translational modifications have been identified, including dual phosphorylation on Thr-180 and Tyr-182 by MAP2K3/MKK3, MAP2K4/MKK4, and MAP2K6/MKK6. Dual phosphorylation can also be mediated by TAB1-mediated autophosphorylation. TCR engagement in T-cells also leads to Tyr-323 phosphorylation by ZAP70. It is dephosphorylated and inactivated by DUPS1, DUSP10, DUSP16, and PPM1D [34]. It is acetylated at Lys-53 and Lys-152 by KAT2B and EP300. Acetylation at Lys-53 increases the affinity for ATP and enhances kinase activity. Lys-53 and Lys-152 are deacetylated by HDAC3. Ubiquitination occurs at Lys15, Lys45, Lys139, Lys152, Lys165, and Lys233 [35]. Five isoforms produced by alternative splicing have been reported: Q16539-1 (considered the canonical) to Q16539-5, with variant 4 being shorter (307 residues) [36,37,38,39]. The specific activation of isoform Q16539-3 is measured based on mitogen stimulation and oxidative stress, while isoform Q16539-4 may play a role in the early initiation of apoptosis [40]. Its subcellular localization is in the cytosol and nucleus [41], with tissue expression occurring predominantly in the brain, heart, placenta, pancreas, and skeletal muscle, and less expression in lung, liver, and kidney (Expression Atlas (Q16539)).

*MAPK11*(p38β): This gene is located on the long arm of chromosome 22 (22q13.13) (GRCh37/hg19 by Entrez Gene) and consists of 7055 base pairs in a minus-strand orientation. It encodes a protein of 364 amino acids (MW = 41,357 Da) (NCBI Entrez Gene 5600). In its sequence, it contains a protein kinase domain at residues 24–308 and a TXY phosphorylation motif at positions 180–182 (Figure 1). There are two ATP binding sites at positions 30–38 and 53. Different post-translational modifications have been identified, such as dual phosphorylation on Thr-180 and Tyr-182 by MAP2K3/MKK3, MAP2K4/MKK4, and MAP2K6/MKK6 [42].

Two isoforms produced by alternative splicing have been reported: Q15759-1, considered the canonical, and Q15759-3, which is the shortest (213 residues) [43]. Its subcellular localization is in the cytosol and nucleus, with tissue expression occurring predominantly in the brain, heart, placenta, lung, liver, pancreas, kidney, and skeletal muscle (Q15759).

*MAPK12* (p38γ): This gene is located on the long arm of chromosome 22 (22q13.33) (GRCh38/hg38 by Entrez Gene) and consists of 16,267 base pairs in a minus-strand orientation. It encodes a protein of 367 amino acids (MW = 41,940 Da) (NCBI Entrez Gene P53778). In its sequence, it contains a protein kinase domain at residues 27–311 and a TXY phosphorylation motif at positions 183–185. There are two ATP-binding sites at positions 33–41 and 56 and one active site at residue 153 (Figure 1). Post-translational modifications have been identified, including dual phosphorylation on Thr-183 and Tyr-185 by MAP2Ks MAP2K3/MKK3 and MAP2K6/MKK6, and ubiquitination for its degradation has been reported [22]. Two isoforms produced by alternative splicing have been reported: P53778-1, considered the canonical, and P53778-2, which is the shortest (357 residues, lacking residues 142–151) [44] (Figure 1). Its subcellular location is in the cytosol, nucleus, and mitochondria, with tissue expression predominantly occurring in skeletal muscle and the heart [45,46].

*MAPK13* (P38δ): This gene is located on the short arm of chromosome 6 (6p21.31) (GRCh38/hg38 by Entrez Gene) and consists of 16,716 base pairs in a plus-strand orientation. It encodes a protein of 365 amino acids (MW = 42,090 Da) (NCBI Entrez Gene O15264). In its sequence, it contains a protein kinase domain at residues 25–308 and a TXY phosphorylation motif at positions 180–182. There are two ATP-binding sites at positions 31–39 and 54. Post-translational modifications have been identified, including dual phosphorylation on Thr-180 and Tyr-182 by MAP2K3/MKK3, MAP2K4/MKK4, MAP2K6/MKK6, and MAP2K7/MKK7. Two isoforms produced by alternative splicing have been reported: O15264-1, considered the canonical, and O15264-2, which is the shortest (257 residues, lacking residues 257–364) [47] (Figure 1). Its subcellular location is in the cytoplasm, cytosol, and nucleus, with tissue expression occurring predominantly in the testis, pancreas, small intestine, lung, and kidney.

### 2.2. p38 Regulation

#### 2.2.1. Transcription Factors in the Regulation of p38

As previously mentioned, the regulation of p38 can occur due to various conditions, kinases, and transcription factors [15]. Zarubin et al. reported that the regulation of p38 can occur due to extracellular stimuli such as osmoregulation, heat, ultraviolet light, inflammatory cytokines such as TNF, growth factors such as CSF-1, or kinases such as MKK3 and MKK6, among other factors [10]. However, the participation of transcription factors is still a growing subject of study. Previously, a series of factors was suggested that might have a regulatory role, such as ATF-1 and 2, Elk-1, SRF, and CHOP-10, among others [48]. Downstream, the regulation of transcription factors such as YY1 and BCL6 by p38 has been reported [49], and several authors have referred to MEF2 as a p38 regulation target [50,51].

In this review, we analyzed the possible participation of transcription factors in the regulation of p38 using bioinformatic tools. Although they are not involved in the main mechanism of kinase regulation, they may be an important part of generating this at the transcriptional level. They may explain why the regulation of certain transcription factors affect the expression of p38, which becomes important for therapeutic purposes. We analyzed the promoter regions of the four isoforms of p38, namely, p38α (MAPK14), p38β (MAPK11), p38γ (MAPK12), and p38δ (MAPK13) [52], using the “search motif” tool of the EPD database (available at https://epd.epfl.ch/; accessed on 17 June 2022). The region from −1000 to 100 relative to the transcription start site of promoters ID MAPK14_1, MAPK11_1, MAPK12_1, and MAPK13_1 was analyzed, which, according to Ensembl (https://www.ensembl.org, 20 June 2022), corresponds to p38, with a cut-off value of p = 0.001.

Transcription factors with at least one binding site in the mentioned region are listed in Table 1, including a selection of those that have been reported as relevant in hematological malignancies.

The assay allows us to assume that the transcriptional regulation of p38 could be related to several malignant processes, given the nature of the transcription factors involved; however, it is necessary to experimentally validate the participation of each transcription factor. Recognizing that further research is required, here, we describe some reported findings related to the participation of these transcription factors in the regulation of p38.

#### 2.2.2. Non-Coding RNAs in the Regulation of p38

Interestingly, and contrary to what is observed with transcription factors, the regulation of p38 by microRNAs has been studied in various pathologies.

Studies indicate that long non-coding RNAs (lncRNAs) play an important role in the pathophysiology of MM [53]. Recent studies have shown that prostate cancer-associated ncRNA transcript 1 (PCAT-1) plays an important role in the pathophysiology of MM and has a regulatory role in p38, where it induces an increase that correlates with proliferation and chemoresistance [54]. Therefore, regulating PCAT-1 could be an important therapeutic target in the context of regulating p38 in MM. Recent studies of glioblastoma have shown that the LncRNA small nucleolar RNA host gene 5 (SNHG5) promotes the p38 protein and induces its activation through phosphorylation [55].

p53-induced noncoding transcript (PINT) is a long intergenic non-coding RNA (linc-RNA). A recent study revealed that nuclear PINT increases the gene expression of the MAPKinase pathways [56]. Since p38 is related to treatment responses in MM, PINT likely increases p38 expression in MM cells and may activate a MKK6/p38 signaling kinase stress cascade in MM patients. Studies have shown that there is significant expression of the miR-106b/25 cluster in MM cells [57]. It is a pro-oncogene, and interestingly, additional studies have shown that this cluster positively regulates p38 activation in MM. Therefore, inhibiting the miR-106b/25 cluster and, as a result, p38, has been considered as a therapeutic alternative in MM [58].

In this review, we used the miRtarbase (https://mirtarbase.cuhk.edu.cn/, 19 July 2022) to determine some of the miRNAs that probably regulate p38 (Table 2). The following microRNAs with a possible role in the regulation of MAPK14 were found: hsa-miR-124-3p, hsa-miR-24-3p, hsa-miR-199a-3p, hsa-miR-200a-3p, hsa-miR-141-3p, hsa-miR-125b-5p, hsa-miR-214-3p, hsa-miR-155-5p, hsa-miR-17-5p, and hsa-miR-106a-5p. The microRNAs found for MAPK11 were hsa-miR-122-5p, hsa-miR-124-3p, and hsa-let-7a-5p, and those found for MAPK13 were hsa-miR-18a-5p, hsa-miR-150-5p, hsa-miR-18b-5p, hsa-miR-3134, hsa-miR-3691-5p, hsa-miR-4434, hsa-miR-4516, hsa-miR-4525, hsa-miR-4531, hsa-miR-4534, hsa-miR-4690-3p, hsa-miR-4731-5p, hsa-miR-4735-3p, hsa-miR-4761-5p, hsa-miR-4773, hsa-miR-5010-5p, hsa-miR-5187-5p, hsa-miR-5589-5p, hsa-miR-5685, hsa-miR-5703, hsa-miR-6778-3p, hsa-miR-6795-5p, hsa-miR-6798-5p, hsa-miR-6814-5p, hsa-miR-6887-5p, and hsa-miR-8082. No miRNAs were found for the isoform MAPK12.

To further confirm the correlation of the miRNAs/p38 axis, we used a bio-predictive website (http://www.targetscan.org/vert_72/, 22 July 2022) (Table 2). The use of TargetScan at conserved sites found possible binding involvement of the miRNAs hsa-miR-3681-3p, hsa-miR-128-3p, and has-miR-216-3p for the isoform MAPK14, while no conserved sites were found for MAPK13. According to TargetScan there are three conserved miRNA sites for MAPK12: hsa-miR-125a-5p, has-miR-125b-5p, and has-miR-4319. Finally, only one conserved site was found for MAPK11: hsa-miR-325-3p.

Studies of gastric adenocarcinoma showed that miR-141 could activate the p38 signaling pathway [59]. Additionally, studies reported that miR-141 could modulate the response to oxidative stress and stimulate tumor growth in mouse models through direct regulation of p38α, which restricted tumorigenesis by blocking proliferation and promoting apoptosis [60,61]. Therefore, this regulatory capacity of miR-141 has been highlighted as a central factor in the response to chemotherapy [62]. It probably plays an important role in MM and chemoresistance, which is a field that offers opportunities to study it and its possible use as a therapeutic target based on its ability to regulate p38.

This analysis and many previous works allow us to establish that multiple miRNAs can regulate the expression of p38 and function in the growth, differentiation, apoptosis, and metastasis of tumors. Given the confirmed involvement of miRNAs in the regulation of p38, these miRNAs may serve as important regulators in various pathologies, including MM, and can be proposed as important therapeutic targets.

### 2.3. p38 Chemicals Inhibitors on Cancer

Despite the evidence, the role and expression of p38 in tumorigenesis is controversial, since it is considered that it may have a dual oncogene or antitumor role. Therefore, different studies have proposed p38 as an important target for cancer therapy. In general, the expression level of p38 in some tumors is high, such as in hematological neoplasms and others such as breast, ovarian, colorectal, and gastrointestinal cancer [63,64,65]. In the search for specific compounds capable of inhibiting p38, many of these have been developed and are undergoing clinical studies, both in cancer and other diseases. For example, there are pyridine imidazoles, such as SB203580, which emerged as inhibitors of proinflammatory cytokines, and their function was later attributed to inhibiting the catalytic activity of p38 through competitive binding in the ATP pocket [66].

SB203580 is a specific inhibitor of p38α [67], although other pharmacological studies demonstrate that these inhibitors inhibit p38α/β but have no effect on p38γ/δ [68]. Studies have shown that this inhibitor can inhibit tumor metastasis and invasion in murine models, by inhibiting the expression of the E-cadherin protein, during gastrulation [69]. In the use of arsenic trioxide (ATO) for therapy in MM, high resistance to treatment has been reported, which is attributed in part to the activation of p38 mediated by ATO in cell lines and primary cultures of MM. In vitro studies using the p38 inhibitor SB203580 increase the inhibition of growth and of the induction of apoptosis through treatment with ATO, as well as the inhibition of IL-6 secretion [70]. Therefore, the use of a combination of ATO and SB203580 has been proposed as an interesting therapeutic alternative in patients with MM. Additionally, as previously mentioned, Syk inhibitors, such as BAY61-3606, R406 or Piceatannol, have been used in the treatment of MM in vitro, showing inhibition of the proliferation and induction of apoptosis mediated by the inhibition of MAPKs such as ERK and p38 [71]. Therefore, the combined use of SB203580 with the Syk inhibitor significantly increases the induction of apoptosis in MM cells. Carfilzomib, a second-generation proteasome inhibitor approved for the treatment of multiple myeloma, shows an important effect in the treatment of osteosarcoma; however, some resistance has developed. Studies show that Carfilzomib can induce p38 activation, and this mediates resistance [72]. SB203580 is a small molecule originally designed as an anti-inflammatory and has been widely used in studies against various types of cancer alone or in combination with drugs [73,74,75]; its toxicity is minimal, and clinical studies continue to asses its application as a neuroprotectant [76] and to prevent post-operative tissue adhesion [77].

VX-745 is another p38 inhibitor; it was developed in 1998 [78], and it was used in clinical studies in the treatment of rheumatoid arthritis. Studies have also shown that VX-745 can inhibit the proliferation of MM cells by inhibiting the secretion of IL-6 by bone marrow stromal cells (BMSCs) and by MM cells, preventing their adhesion, implicated as a probable mechanism of chemo resistance [79]. VX745 was first described in 1998, and in 1999, a clinical trial was initiated for the treatment of rheumatoid arthritis [80]. Adverse effects were observed in the CNS, so another p38α inhibitor (VX702) was developed that could not cross the blood–brain barrier. But its study continued in patients with dementia, and its study is currently continuing in clinical phases for the treatment of Alzheimer’s disease and dementia with Lewy bodies. Variants of VX745 have recently been developed and used in several diseases; these variables are quite similar to VX745, but the pharmacokinetics have been improved [81].

8-NH2-Ado is a nucleoside analogue that has been used in the treatment of hematological malignancies such as MM, but still has not clearly exhibited effectiveness in the clinic. However, in vitro, it has been shown that 8-NH2-Ado has significant cytotoxic activity [82,83]. Recent studies have shown that the mechanism of action of 8-NH2-Ado, at least in MM, is the inhibition of the phosphorylation of many proteins, such as ERK1/2, Akt and p38, and because of this, the inhibition of the phosphorylation of these proteins is the induction of apoptosis in these MM cell lines treated with 8-NH2-Ado [83].

SCIO-469, is a selective and active ATP-competitive p38α inhibitor that has been tested as monotherapy or in combination with bortezomib in relapsed MM, as well as in a murine model of MM, with excellent results [84,85], and even reduces the development of osteolytic bone lesions in this MM model [86]. Additional studies confirm that SCIO-469 treatment can suppress factors in the bone marrow microenvironment to inhibit MM cell proliferation and adhesion and alleviate osteolytic activation in MM [87]. Phase II clinical studies have shown that SCIO-469, as a monotherapy or in combination with bortezomib, was well tolerated in patients with relapsed refractory multiple myeloma [88], as was also the case for in patients with myelodysplastic syndrome [88].

BIRB-796 is a pan-p38 inhibitor with in vitro activity for p38α/β/γ/δ. It has been shown that treatment of MM with bortezomib induces p38 activation and that the inhibition of p38 reverses resistance to bortezomib-dexamethasone or 17-AAG. Studies show that the use of BIRB-796 increases cytotoxicity and caspase activation, induced by the treatment [89]. Thus, also in BMSC, BIRB-796 inhibits the secretion of IL-6, VEGF, TNF-α, and TGF-β1. BIRB-796 also inhibits IL-6 secretion induced in BMSCs through adherence to MM cells, thereby inhibiting tumor cell proliferation. Therefore, it is suggested that BIRB-796 reverses chemoresistance in the BM microenvironment, offering an important clinical alternative, alone and in combination with conventional chemotherapy, to improve patient outcomes in MM [89]. Studies show that BIRB-796 has a greater affinity for p38δ and that BIRB-796 and its variables, such as E1, decrease tumor volume and show acceptable characteristics in in vivo pharmacokinetic experiments, and do not show any toxicity [81,90].

LY2228820 is a selective inhibitor of p38α and p38β and was created for use in cancer [91], where it was shown to significantly inhibit tumor proliferation in in vitro and in vivo models of melanoma, non-small-cell lung cancer, ovarian cancer, glioma, myeloma, and breast cancer. However, recent studies have shown that it can inhibit the expression of EGFR, independent of p38 [92]. Recent studies have shown that DK7 (THZ1) inhibitors can inhibit the expression of p38 and induce tumor shrinkage; however, the use of the combination of THZ1 and LY2228820 has a synergistic effect on inhibiting the proliferation of cancer cells. [93]. In MM, as with other p38 inhibitors, the use of LY2228820 exhibited a modest cytotoxic effect on tumor cells, but increased the tumor cytotoxicity of bortezomib and inhibited the secretion of IL-6 from BM stromal cells and BM mononuclear cells (BMMNC) derived from MM patients in remission, and significantly inhibited osteoclastogenesis in vitro and in vivo in a xenographic model of human MM [94]. This suggests that LY2228820 could be a new therapeutic alternative to improve outcomes in patients with MM, both by improving the effect of bortezomib and by reducing osteoskeletal events. In clinical studies in advanced cancer patients, LY2228820 showed good results in tumor shrinkage with acceptable side effects at the minimum dose used. Treatment-related AEs were grade 1/2 with a treatment-related safety profile consisting primarily of rash, fatigue, constipation, and nausea. At the maximum dose explored, DLTs of grade 3 ataxia and grade 2 dizziness were reported [95].

SD-169 is an ATP-competitive inhibitor of p38α and also weakly inhibits p38β. Studies have shown that p38 is capable of activating osteoclasts and bone resorption and, on the other hand, inhibits osteoblast and bone formation, both of which result in bone destruction in MM. In murine models of MM, it has been shown that the inhibition of p38 with SD-169 decreases the ability of myeloma cells to cause bone destruction in vivo. Therefore, the use of SD-169 could be an effective treatment to treat tumor-induced osteolytic bone lesions in patients with myeloma [96].

It is clear that p38 expression, and especially the p38α/β isoforms, plays an important role in the development and progression of multiple tumors, including MM. Therefore, the use of chemical p38 inhibitors specifically designed around isoforms α/β is of great relevance as therapeutic alternatives in patients with MM.

## 3. p38 Expression in Multiple Myeloma

The participation of p38 in several cellular processes in the context of multiple myeloma has been reported. p38 is constitutively activated in myeloma and plays a fundamental role in bone destruction in this type of cancer, probably due to its regulatory effect on DKK-1 and MCP-1. Even attenuated p38 expression is sufficient to reduce lesions in bone in vivo, so it can be proposed as a potential therapeutic target for the treatment of bone lesions in myeloma [11]. This observation is consistent with those mentioned above using a p38 inhibitor, VX-745 [79]. Additional studies using the p38 inhibitor 8-NH2-Ado, whose effects are described above, once again demonstrate the importance of the role of p38 in MM [83].

As already mentioned, p38 activation plays an important role in resistance to cytotoxic chemotherapeutic drugs in the treatment of MM. Studies on the roles played by the isoforms of p38 have revealed that the knocked down p38α isoform is more sensitive to bortezomib or arsenic trioxide treatment, while the knocked down p38β/γ isoform is more resistant to the same treatment, which suggests that p38α plays a more important role in resistance, while p38β/γ is not as relevant in this function [97]. Interestingly, knocking down p38δ shows a significant effect on proliferation without treatment, which is affected under treatment conditions. This suggests that the p38δ isoform plays an important role in the proliferation of MM cells. This is probably due to the phosphorylation capacity shown by p38δ on Erk1/2, while p38α has a greater activation effect on AKT and IKK, as well as the NFκB pathway. This would explain the differential roles of p38 isoforms in MM cells.

In order to find studies reporting the clinical importance of p38 expression in MM, the public database Gene Expression Omnibus (GEO), NCBI, (https://www.ncbi.nlm.nih.gov/geo/query/acc.cgi?acc=GSE146649, 18 August 2022) was searched by using the words “multiple myeloma” and “healthy donors” and selecting “homo sapiens” in the organism filter and “expression profiling by array” as the study type. Finally, the results were manually curated to identify the information of the datasets shown. We selected the GSE146649 dataset entitled “Expression data from bone marrow mesenchymal stromal cells obtained from healthy donors and myeloma patients”, and we performed an analysis of the p38 isoforms relative expression reported in the dataset. (Figure 2). Despite robust experimental evidence, when we compare the relative expression of isoforms MAPK11 and MAPK12, the expression at the diagnostic (DX) stage is decreased in comparison to the healthy donor (HD), complete response (CR), and early relapse (ER) stages (** *p* < 0.01 *** *p* < 0.001). On the other hand, isoform MAPK14 shows a decrease in relative expression in the DX stage, but it is only significant vs. the CR stage (* *p* < 0.05). MAPK13 shows no significant difference between stages. This analysis does not show specific results on the expression of each of the isoforms in MM and the different stages, probably due to the limitation in the sample size of each of the analyses. However, it allows us to suggest that, at least for the p38α (MAPK 14) and p38β (MAPK11) isoforms, an increase in expression is observed after treatment and is maintained in the complete response and early relapse stages. This is consistent with what has been reported in vitro trials where the treatment induces an increase in the expression of p38 α/β, which can be favored clinically in combination with chemical inhibitors of p38 α/β. These results suggest a potential role of p38 in the development of MM; however, as mentioned above, there are few clinical studies on this topic. What this suggests, and as has been previously reported [79], treatment induces p38 expression in MM, and this is maintained in patients who respond to treatment. Obviously, since it is a MAP kinase, rather than its expression, its activation or biological activity must be evaluated, which could not be achieved in these reviewed studies. Therefore, it is very important to clinically evaluate the activation of the different p38 isoforms.

## 4. p38 as a Molecular Target in Multiple Myeloma Therapy

Multiple studies have shown that interleukin-6 (IL-6) is an important growth factor involved in the physiopathogenesis of multiple myeloma [98], and through an autocrine/paracrine form, promotes the survival and proliferation of myeloma cells. It has also been shown that growth factors such as granulocyte colony-stimulating factor (G-CSF), as well as IL-10, may also play a role in MM cell lines [99,100]. IL-6 can induce the activation of different signaling pathways that may be involved in the physiopathogenesis of this disease. As already mentioned, these include the Jak-Stat, Ras/Raf/Mek/Erk, and AKT pathways [99], which are essential for MM cell proliferation. Additionally, it has been shown that mutations in Ras (N-Ras and K-Ras) lead to constitutive activation that also influences a more malignant phenotype in patients with MM [101]. This Ras activation-dependent proliferation may be independent of the Mek/Erk pathway, which makes the use of inhibitors of this pathway unfeasible as a therapeutic alternative in cases of MM with malignant phenotypes [102]. Studies have shown that heme oxygenase-1 (HO-1), an enzyme that provides potent cytoprotection, cell proliferation, and drug resistance [103], is expressed and correlates with the expression of IL-6 in the bone marrow microenvironment of MM patients and autocrine IL-6 in MM patient cells, and both HO-1 and IL-6 have been associated with disease severity in MM [104]. The increased expression of HO-1 can induce the increased expression of IL-6 through p38 and other MAPKs. Therefore, the chemical inhibition of p38 significantly inhibits the expression of IL-6 mediated by HO-1, and this may have therapeutic importance in MM. Other growth factors, such as insulin-like growth factor 1 (IGF-1), have also been shown to induce activation of the Akt and Mek/Erk pathways in MM cell lines, where it appears that the PI-3/Akt pathway plays a major role [99,100].

As already mentioned in the section on chemical inhibitors for p38, certain studies have associated the activity of the p38 pathway with the pathophysiology of MM, since the inhibition of this pathway by the chemical inhibitor VX-745 induces inhibited IL-6 secretion in MM cells and inhibited cell proliferation. Thus, the use of a combination of bortezomib with SCIO-469 or BIRB-796, as well LY2228820, also inhibits osteoclastogenesis [12,89,94].

Therefore, it is believed that p38 indirectly participates in the physiopathogenesis of MM in an indirect way, and this inhibition may have therapeutic implications in the treatment of MM [12]. Studies have shown that treatment with bortezomib induces the induction of apoptosis through p38 inhibition, which is mediated by the increased expression of p53 and decreased expression of Bcl-XL and Mcl-1 [85]. It has also been reported that p38 inhibition decreases the expression of IL-6 and VGEF by BM stromal cells, resulting in the inhibition of MM cell proliferation and adhesion [87], thus decreasing the tumor burden and angiogenesis in murine models of MM [84,86].

The interaction between bone marrow stromal cells (BMSCs) and multiple myeloma cells is very important in the pathogenesis of MM through the secretion of growth factors, cytokines, and extracellular vesicles. Exosomes appear to play an important role in the communication between BMSCs and MM cells through the transfer of cytokines between other molecules. In an in vivo model, it was shown that BMSC and MM cells could exchange exosomes carrying certain cytokines, inducing increased growth of MM cells and resistance to bortezomib [105]. Thus, they also induce the activation of several important pathways for survival, including p38. Therefore, these studies suggest that p38 inhibitors may have an important therapeutic role in preventing the effect of exosome-mediated BMSCs on the proliferation, migration, survival, and drug resistance of MM cells.

Recent studies have shown that the TAK1 inhibitor induces the inhibition of proliferation and apoptosis in MM cells through the constitutive or melphalan-mediated inhibition of TAK1, NF-kB, and p38 [106]. Additional studies have shown that the selective TAK1 inhibitor 5Z-7-oxozeaene shows synergistic potential with bortezomib, inducing increased inhibition of proliferation and apoptosis [107]. This biological activity is related to the inhibition of TAK1, which induces the inhibition of JNK, MAPK p38, and Erk, which are activated by TAK1 in MM cell lines. Therefore, various studies have reported the use of chemical inhibitors of growth signaling pathways in MM as a therapeutic alternative, such as inhibitors of the Stat3- and Erk2-dependent IL-6 activation pathways. Another candidate is the proteasome inhibitor PS-341, which has been shown to activate the JNK pathway and inhibit the Erk1/Erk2-dependent pathway [79]. The possibility that MAPK pathways participate in the growth and physiopathogenesis of MM in general has led to the development of studies with the aim of using MAPK inhibitors as potential therapeutic agents alone or in combination with chemotherapy or other conventional or novel therapies.

MAPKAPK2 (MK2) is the direct substrate of MAPK p38. Recent studies have shown that MK2 plays an important role in the pathophysiology of MM [108]. These studies suggest that using p38 inhibitors that affect MK2 or direct MK2 inhibitors could be an important therapeutic alternative for MM. One study reported that rafoxanide, an antiparasitic, is capable of inducing mitochondria-dependent proliferation inhibition and apoptosis in MM cells [109]. This is a consequence of the inhibition of p38 and Stat-1. Therefore, non-specific inhibitors that impact the p38 pathway may also be therapeutic alternatives for MM.

It has recently been reported that p38 also regulates the expression of the NKG2D and DNAM-1 ligands in MM cells in a drug-dependent manner, sensitizing them to the induction of death by NK cells [110]. This has been determined to be mediated by the p38 activation of transcription factor E2F1. These results suggest that p38 inhibition could be a therapeutic alternative in cases of immunoresistance in MM mediated by NK cells.

Spleen tyrosine kinase (Syk) is an intracellular enzyme that plays an important role in the activation of B cells or T cell receptors. Studies have shown that Syk inhibitors can inhibit proliferation and induce apoptosis in these kinase cells in MM, where inhibition of the p38 pathway has been shown to be one of the consequences of such treatment [71]. The combination of the Syk inhibitor and p38 inhibitor results in the increased induction of apoptosis in MM cell lines.

Trifluoperazine is a drug used in psychosis, but recent studies have shown that it inhibits tumor growth in different types of cancer, including MM cells [111]. Additional studies found that this drug in combination with bortezomib had a cytotoxic effect in in vitro and in vivo models of MM by inducing proliferation inhibition and apoptosis mediated by p38 inhibition [111].

In vivo and in vitro studies have shown that celastrol, a pentacyclic nortriterpen quinone, has an antitumor effect in MM and other types of cancer [112]. Recent studies have shown that celastrol induces apoptosis in MM cell lines alone or in combination with bortezomib, and that the cytotoxic effect is also associated with inhibition of the IRAK4/ERK/p38 pathway [113].

Tris(dibenzylideneacetone)dipalladium (Tris DBA) is a small molecule of the palladium complex that induces inhibited cell growth and proliferation in multiple myeloma cells, either alone or in combination with bortezomib. This effect is a consequence of the inhibition of downstream p38 signaling [114]. Interestingly, Tris DBA reverses hypoxia-mediated drug resistance by inhibiting p38 and Hif-1α [115].

The cell-derived protein kinase T-LAK/PDZ-binding kinase (TOPK/PBK) has been proposed as a potential therapeutic target due to its low expression in most normal tissues and high expression in various tumors, including MM. Recent studies have reported that OTS514, a TOPK/PBK inhibitor, had a cytotoxic effect on MM cell lines [116]. OTS514 induces apoptosis by inhibiting FOXM1, Akt, and p38.

Histone deacetylases are potential therapeutic targets in hematological malignancies. Recent studies have shown that a new histone deacetylase inhibitor induces cytotoxicity in MM cells via apoptosis and was able to induce cytotoxicity in myeloma cells co-cultured with bone mesenchymal stromal cells and osteoclasts previously treated with the inhibitor [117]. This inhibitor was found to suppress osteoclast differentiation and resorption in vitro by inhibiting ERK, p38, AKT, and JNK, which prevented MM-associated bone loss in an in vivo model. These results support the idea that inhibitors of MAPKs, and in particular, p38, may have potential clinical use in multiple myeloma treatment in the near future.

Various groups have shown that p38 is constitutively active in myeloma cells and that this leads to osteolytic bone destruction [11,79,87]. Therefore, treatment with a p38 inhibitor decreased tumor burden and bone lesions in a murine myeloma model, prolonging survival [86]. Additionally, the inhibition of p38 has an important effect on reducing osteolytic bone lesions induced by MM, reducing osteoclastogenesis and improving osteoblastogenesis [11]. The important role of p38 in bone damage induced by MM was confirmed using shRNA specific to p38α in vitro, where lower bone resorption associated with the knockdown effect of p38 was observed. Therefore, p38 inhibition in special p38α/β inhibitors is positioned as an important therapeutic alternative in the prevention of osteolytic bone lesions caused by MM [11,118].

The CXCR4 chemokine receptor is expressed in a wide variety of hematological malignancies, including MM [119], and in conjunction with its ligand SDF-1, it plays an important role in cancer progression. Studies in which researchers generated a humanized mAb, hz515H7, which binds to human CXCR4, showed that this binding inhibits the signaling pathway induced by SDF-1, reducing the phosphorylation of Akt, Erk1/2, and p38, which strongly inhibits cell migration and proliferation [120]. Therefore, the inhibition of the p38 signaling pathway as part of the mechanism of action of humanized mAbs used therapeutically in MM may be important in their therapeutic efficacy. Additionally, studies using gambogic acid (GA), a xanthone that inhibits CXCR4 signaling, demonstrated that it is capable of suppressing osteoclastogenesis induced by MM cells [121], inhibiting activation of the NF-kB factor transcription pathway, which regulates CXCR4 expression. GA was found to suppress the SDF-1α-induced chemotaxis of MM cells and signaling pathways downstream of CXCR4 and inhibit Akt, p38, and Erk1/2 activation. Interestingly, the MM-mediated suppression of osteolytic bone damage is regulated by IL-6 inhibition and consequent p38 inhibition. These findings suggest that modulating chemotaxis factors and cytokines such as CXCR4 and IL-6, the most common factors of which include activation of the p38 signaling pathway, may be important therapeutic targets, through bifunctional alternatives or directed toward the inhibition of p38 in MM.

Studies have reported that the mTOR signaling pathway is involved in the pathophysiology of MM [122], and a small mTOR inhibitory molecule (SC06) was found to induce the inhibition of tumor growth in an in vivo model of MM [123]. This inhibition did not affect other signaling pathways, such as AKT, ERK, c-Src, and JNK, but showed a significant effect on p38 in some of the analyzed MM cell lines. This suggests that the activation of the p38 signaling pathway in MM may be mediated in part by the mTOR pathway, and that the cytotoxic effect from the inhibition of the mTOR signaling pathway may be, among other pathways, the inhibition of the p38 pathway.

Studies have shown that MM cell lines can express high levels of TLR5, which, when activated with its specific ligand, flagellin, induces IL-6 expression through the activation of NF-κB, which is, in turn, activated by p38 and PI3K/AKT pathway signaling. This leads to greater resistance to chemotherapeutic agents [124]. In addition, MM cells have also been shown to express TLR7 and TLR9 [125], which also induce the expression of IL-6 and, by activating the p38 pathway, induce chemoresistance [126]. These results suggest that there are mechanisms of the innate immune system that may favor the development and chemoresistance of MM cells, which, by activating survival signaling pathways such as p38, may play a role in the pathophysiology of MM. Thus, once again, inhibition of the p38 pathway is emerging as an important therapeutic target.

All-trans retinoic acid (ATRA) has been used in the treatment of multiple myeloma, but ATRA-induced chemoresistance has been reported in myeloma patients. This is associated with the induction of apurinic endonuclease/redox factor-1 (Ape/Ref-1) expression, which leads to MDR transactivation [127,128]. Studies have revealed that ATRA activates the p38 pathway and can promote Ape/Ref-1 expression, and this was reversed by treatment with a chemical inhibitor of p38. These studies suggest that p38 has a role in chemoresistance to ATRA treatment; thus, it may be an important target in ATRA-mediated chemoresistance.

Finally, as was previously discussed, treatment with arsenic trioxide (ATO) promotes p38 activation in MM cell lines, while treatment with ATO in combination with a p38 inhibitor abolished resistance to ATO, so it was suggested that p38 could be involved in the promotion of ATO chemoresistance in MM cell lines [70].

All these data reinforce the idea that p38 activity, especially p38 α/β, plays an important role in the pathophysiology of MM and that it directly or indirectly intervenes in the processes of chemoresistance, proliferation, and growth in MM. Additionally, it plays a role in the physiopathogenesis of MM. Therefore, its therapeutic intervention in MM continues to be a valid strategy, whether through direct inhibition or by altering any of the mechanisms that regulate its expression and activation (Figure 3).

## 5. Conclusions

Many research papers have highlighted the importance of MAPKs in the physiopathogenesis of hematological malignancies and have shown that they have important roles in the regulation of the growth and apoptosis of these hematological cells. In the specific case of MM, it has been described that the Raf/Mek/Erk pathway undoubtedly participates in its pathophysiology, contributing to its growth, while the JNK and p38 pathways seem to have an indirect role in growth or the mechanisms of resistance to current therapies.

Over the past two decades, the possibility of using chemical inhibitors of the MAPK pathways as therapeutic alternatives has been raised, in both in vitro assays and clinical trials. Currently, there are important advances in this area, with important limitations, probably due to the heterogeneity of gene regulation in MM; in addition, a flow cytometry analysis of phosphorylation profiles showed that the activation patterns of signaling pathways such as Stat-3 and MAPK p38 in cells from MM patients and MM cell lines are heterogeneous [129]. However, there are currently important clinical advances in MM treatment with the use of immunomodulators in combination with chemical inhibitors of the signaling pathways that could lead to therapeutic protocols with great potential.

In addition to studies that are currently under development to find new therapeutic alternatives for MM, Mek inhibitors alone or in combination have been used in the treatment of MM, as well as chemical inhibitors of p38α/β in MM patients.

The accumulated evidence on the role of MAPK in hematological malignancies has allowed the development of new therapeutic agents, some of which, as already mentioned, are undergoing clinical trials. But without a doubt, greater knowledge regarding the participation of p38 and its isoforms in the pathophysiology of MM will allow us to significantly define its importance as a therapeutic target.

Studies have shown that the isoforms of p38 have different patterns of expression and biological functions, and efforts are currently focused on understanding the role of these isoforms in malignant neoplasms. In addition to the above, these isoforms present post-translational variants, and it is believed that they may also have a selective role in cancer, specifically in hematological malignancies.

## Figures and Tables

**Figure 1 cancers-16-00256-f001:**
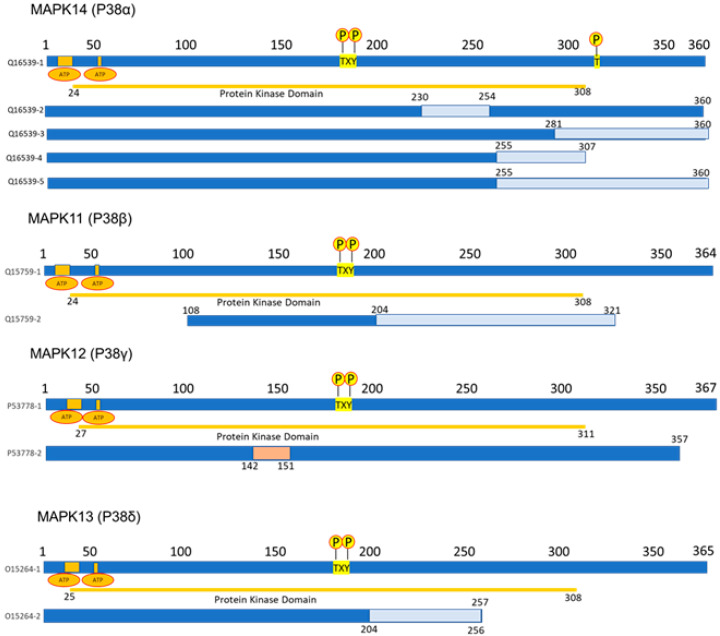
Illustration of the structure of the different p38 isoforms with their corresponding alternative splicing isoforms and their different domains. Solid blue line represents the gene sequence of the canonical isoform for each isoform. Blue line with segments in lighter blue represent the variants of each isoform, where the light blue color represents segments with a different gene sequence to the canonical one. Yellow line indicates the protein kinase domain. Yellow circles with the letter P indicate the TXY phosphorylation motif (p38α reports an additional phosphorylation site by ZAP70 (T 323)). Orange circles indicate ATP-binding sites. Pink line in the p38γ isoform variant P53778-2 represents a lack of residues.

**Figure 2 cancers-16-00256-f002:**
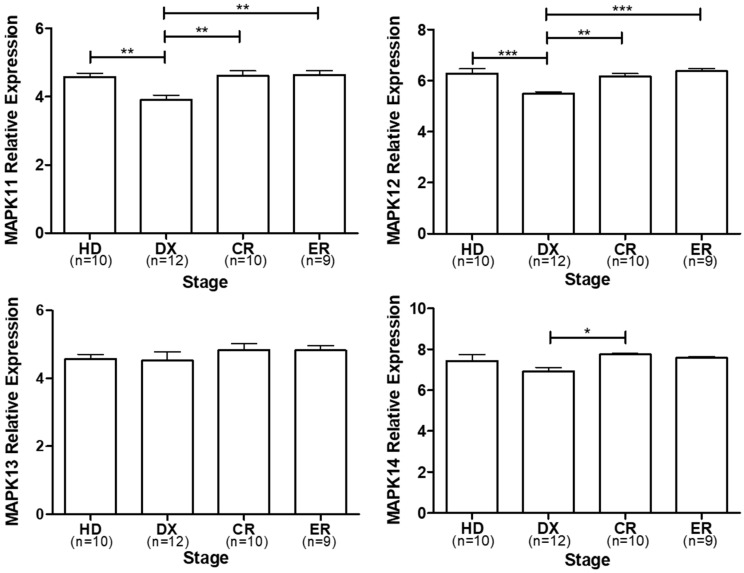
Expression of four isoforms of p38 was evaluated using GEO dataset: GSE146649 at different MM stages. HD, healthy donor; DX, diagnostic; CR, complete response; ER, early relapse (n indicates the number of samples reported and used for this graph) (* *p* < 0.05, ** *p* < 0.01, *** *p* < 0.001). ANOVA was performed with a Bonferroni correction using GraphPad Prism 5.0.

**Figure 3 cancers-16-00256-f003:**
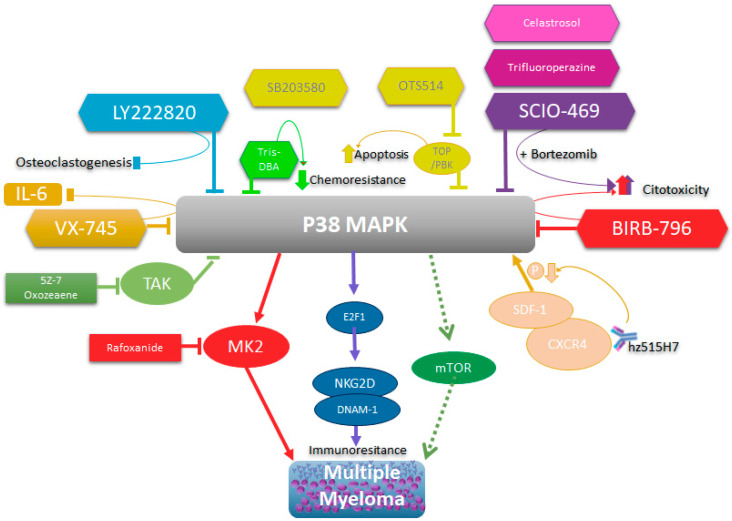
Schematic representation of p38’s role in multiple myeloma and its biological consequences of inhibition.

**Table 1 cancers-16-00256-t001:** Putative binding sites for transcription factors in p38 promoter isoforms in EPD.

Transcription Factor	Putative Binding Site Positions Relative to TSS			
	MAPK14_1	MAPK11_1	MAPK12_1	MAPK13_1
ARNT: HIF1A	−466, −455, −260	−93,	−40, −88	−120, 32
ASCL1	−933, −901, −880, −878, −785, −603, −602, −392, −391, −248, −59, 22, 2	74, 73, −459, −506, −550, −575, −609, −610, −755, −830, −937	44, 42, −2, −3, −190, −214, −215, −309, −310, −410, −411, −518, −523, −666, −688, −810, −812, −882, −883, −968, −982	−993, −810, −809, −551, −278, −264, −254, −185, −184, −103, −102, −87, −40, −17, 25, 26
ATF4	−823, −777, −774			
ATF7	−824, −823		−92	
Ahr: Arnt	−466, −455, −334, −260	−93		−120, 32
Arid3a	−860			−929, −490, −485
Ascl2	−901, −900, −785, −392, −391, −381, −380, −249, −248, −59, 24, 25	−380, −829	−2, −214, −309, −523, −524, −882, −883	−809, −580, −278, −185, −184, −87
Atf1	−823	−579	−92	−697
BACH2	−633, −629	−392		−508, −504
Bcl6	−874	−356, −497, −672, −732, −884	−382, −927	
CEBPA	−979, −823, −549		−548	−947
CTCF	−768, −513, −275, −209, 1	−178, −219, −221, −253, −383, −584, −848	27, −52, −76, −146, −270, −336, −601, −672, −799, −891	−870, −87, −54, −14
E2F1	−597, −596, −270, −269, −218, −217	−183, −184, −290, −543, −544, −660	35, 34	15
EGR1	−465, −394, −28	34, −8, −252, −783, −903, −949	30, −256, −335, −602, −744, −917	−866, −835, −604, −384, −278, −246, −147, −61, −39, −14
ELF1	−62, −28, 33	82, −126, −154, −489, −880	24, −120, −438	−155, 73, 76
JUN	−887, −822			−609, −460
KLF4	−770, −587, −453, −334	−91, −684	−94, −290, −906	−380, −122, −58
MYC	−942, −785, −603, −602, −444, −392, −391, −266	−230, −506	−3, −87, −88, −518, −523, −643, −644	−984, −606, −199, −185, −184, −103, −102, −87
SMAD2:SMAD3: SMAD4	−896, −464, −251, −204	−147, −211, −926	−260, −266, −563, −625, −829, −865, −910	
YY1	−977, −962, −547, −98			−949, −814, −672
XBP1	−943	−578		−738
TP53	−759, −758	−578	−754	−959, −958, −437, −125, −124
Stat4	−419	−672, −676		−395
SP1	−925, −881, −659, −633, −499, −225, −186, −141, −114, 72	63, −12, −33, −46, −87, −141, −165, −192, −200, −279, −294, −347, −508, −527, −543, −598, −650, −683, −759, −905	78, 28, 16, −19, −47, −103, −147, −164, −195, −206, −267, −289, −317, −592, −670, −740, −803, −822, −853, −905	−868, −837, −666, −522, −329, −290, −257, −205, −59, −44, −25, −9
POU1F1	−575	−407		−919, −490, −395
NFKB1	−982, −938, −552, −506, −32	−69, −101, −102, −788, −945	34, 5, −62, −314, −513, −586, −899, −961	−988, −944, −467, −466, −232, −211

**Table 2 cancers-16-00256-t002:** Possible miRNAs regulating isoforms of p38.

	MAPK11	MAPK12	MAPK13	MAPK14
Predicted microRNAs (miRtarBase)	hsa-miR-122-5p, hsa-miR-124-3p, hsa-let-7a-5p		hsa-miR-18a-5p, hsa-miR-150-5p, hsa-miR-18b-5p, hsa-miR-3134, hsa-miR-3691-5p, hsa-miR-4434, hsa-miR-4516, hsa-miR-4525, hsa-miR-4531, hsa-miR-4534, hsa-miR-4690-3p, hsa-miR-4731-5p, hsa-miR-4735-3p, hsa-miR-4761-5p, hsa-miR-4773, hsa-miR-5010-5p, hsa-miR-5187-5p, hsa-miR-5589-5p, hsa-miR-5685, hsa-miR-5703, hsa-miR-6778-3p, hsa-miR-6795-5p, hsa-miR-6798-5p, hsa-miR-6814-5p, hsa-miR-6887-5p, hsa-miR-8082	hsa-miR-124-3p, hsa-miR-24-3p, hsa-miR-199a-3p, hsa-miR-200a-3p, hsa-miR-141-3p, hsa-miR-125b-5p, hsa-miR-214-3p, hsa-miR-155-5p, hsa-miR-17-5p, hsa-miR-106a-5p
Bio-predicted (TargetScan)	hsa-miR-325-3p	hsa-miR-125a-5p, has-miR-125b-5p, has-miR-4319		hsa-miR-3681-3p, hsa-miR-128-3p, has-miR-216-3p

## Data Availability

The data presented in this study are available in this article.

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
