# Peer review of "p38 Molecular Targeting for Next-Generation Multiple Myeloma Therapy"

_cancers, 2024, doi:10.3390/cancers16020256_

Round 1

Reviewer 1 Report (Previous Reviewer 2)

Comments and Suggestions for Authors

This review paper summarized the recent research about the p38 MAPK expression, function, and different isoforms regulation, also the possible clinical application of p38 MAPK signaling pathway in multiple myeloma. My comments are listed below:

1.     Fig. 1 legend needs to improve. What do light blue, ATP and other symbols represent?

2.     Line 320, MPAK-38 means what?

3.     Fig2, add the statistic analysis method in figure legend.

4.     The manuscript structure needs to improve and be more precise according the title.

Author Response

p38 MAPK molecular targeting for next-generation multiple myeloma therapy

Manuscript #. cancers-2759328

 We thank the reviewer for their detailed and helpful comments and suggestions to improve our manuscript. We have answered to each comment, and the point-by-point responses to the reviewer’s concerns are provided below.

Review 1

This review paper summarized the recent research about the p38 MAPK expression, function, and different isoforms regulation, also the possible clinical application of p38 MAPK signaling pathway in multiple myeloma. My comments are listed below:

R: Thank you for your comments and review

  1. 1 legend needs to improve. What do light blue, ATP and other symbols represent?

R= The legend of Figure 1 was modified and the information on the different symbols was added, according to the reviewer's comments.

  1. Line 320, MPAK-38 means what?

R= It was a typing error for MAPK, the word was modified to p38 to be consistent with the rest of the manuscript.

  1. Fig2, add the statistical analysis method in figure legend.

R= The statistical analysis method was added in the legend of Figure 2.

  1. The manuscript structure needs to improve and be more precise according the title.

R= The structure of the manuscript was modified where possible according to the comments of all reviewers.

Reviewer 2 Report (New Reviewer)

Comments and Suggestions for Authors

This extremely long and abundantly documented review paper is proposed as exposing the interest of targeting xp38 in the treatment of multiple myeloma.

In fact, it is a long list of cellular pathways, enzyme genes and structure, potential drugs, in vitro and animal studies, regarding a multitude of cancer types and models. Part 4 is the only one really dealing with multiple myeloma, yet there are many redundancies within this chapter and repetitions from previous parts of the document. All in all, the reader has no clear idea of the way by which p38 isoforms, largely detailed and shown to have very different mechanisms of action, are involved in the pathogenesis of multiple myeloma and why they (especially which one) should be targeted. The two figures shed no light on this question and lack detailed description (especially figure 2). Moreover, dealing with potential drugs for clinical use, toxicities or side effects are barely evoked (in fact just once with a wrong example). A painfully incomplete list of abbreviations is provided at the end of the document, leaving the reader trying to guess what the authors are referring to in too many instances.

Apart from this globally fuzzy impression, crucial information about the way the review was built is completely lacking : data bases interrogated, research keywords, research period, languages, numbers of articles retrieved, supervision, number of articles rejected an why… Some bioinformatic data bases are mentioned and results produced yet with no indication of the tool(s) or protocol(s) used to interrogate them.

On a more formal perspective, the manuscript inexplicably contains paragraphs in different size fonts as well as large highlighted parts, mostly in bright , yellow (very disrespectful to the reader) and in a few other colors at some places.

The reference list is totally haphazard, in a strange global format but with various numbers of authors (often just one !), no homogeneization of title styles (only the first word should be with a capital first letter), useless mention of months and day of publication (although not consistently), mention of recent or older access to some publications (uninformative), some references incomplete… Moreover, less than 20% of the long list of references appeared within the past 3 years.

Comments on the Quality of English Language

Although globally correct, the style of the manuscript is heterogeneous with some sentences lacking verbs and quite a number of grammatical errors.

Author Response

p38 MAPK molecular targeting for next-generation multiple myeloma therapy

Manuscript #. cancers-2759328

 We thank the reviewer for their detailed and helpful comments and suggestions to improve our manuscript. We have answered to each comment, and the point-by-point responses to the reviewer’s concerns are provided below.

Review 2

This extremely long and abundantly documented review paper is proposed as exposing the interest of targeting xp38 in the treatment of multiple myeloma.

R: Thank you for your comments and review

  1. In fact, it is a long list of cellular pathways, enzyme genes and structure, potential drugs, in vitro and animal studies, regarding a multitude of cancer types and models. Part 4 is the only one really dealing with multiple myeloma, yet there are many redundancies within this chapter and repetitions from previous parts of the document. All in all, the reader has no clear idea of the way by which p38 isoforms, largely detailed and shown to have very different mechanisms of action, are involved in the pathogenesis of multiple myeloma and why they (especially which one) should be targeted. The two figures shed no light on this question and lack detailed description (especially figure 2). Moreover, dealing with potential drugs for clinical use, toxicities or side effects are barely evoked (in fact just once with a wrong example). A painfully incomplete list of abbreviations is provided at the end of the document, leaving the reader trying to guess what the authors are referring to in too many instances.

R= Thank you for such important comments and below we will try to respond and make changes to the manuscript as much as possible. As we mentioned in the introduction of the new version (marked in yellow). At first, we addressed general topics of p38, some of its multiple signaling mechanisms, its structural characteristics, as well as its regulation mechanisms mediated by transcription factors or miRNAs. In order to put the molecular characteristics of the p38 signaling pathway in a general context and being consistent with the reviewers' suggestions, a section on the different chemical inhibitors of p38 in the context of cancer in general was added, highlighting the possible use in MM. This is why the probably long introduction and general aspects of p38 and not only in the context of MM. We highlighted where possible which of the isoforms is most important in the context of MM and its importance as a therapeutic target (see sections highlighted in yellow). Figure 2 and its possible importance in the manuscript were described in more detail. The toxicities and side effects of chemical inhibitors used in clinical studies were described where possible. The list of abbreviations was adjusted and modified.

  1. Apart from this globally fuzzy impression, crucial information about the way the review was built is completely lacking: data bases interrogated, research keywords, research period, languages, numbers of articles retrieved, supervision, number of articles rejected and why… Some bioinformatic data bases are mentioned and results produced yet with no indication of the tool(s) or protocol(s) used to interrogate them.

R= In order to clarify the order and purpose of the manuscript, the order and way in which the manuscript was constructed was mentioned in the introduction (marked in yellow). The source information of the bioinformatics studies was detailed (highlighted in yellow)

  1. On a more formal perspective, the manuscript inexplicably contains paragraphs in different size fonts as well as large highlighted parts, mostly in bright, yellow (very disrespectful to the reader) and in a few other colors at some places.

R= Thank you for your comments and observations, these changes in font sizes were caused by the change in format when sending the manuscript in the journal's template. In the original Word file, it appears correct. The paragraphs highlighted in yellow were changes and additions in response to the previously suggested revisions. In this version color have been removed and only the new changes in this revision are highlighted in yellow. The rest of the highlighted colors were removed.

  1. The reference list is totally haphazard, in a strange global format but with various numbers of authors (often just one!), no homogenization of title styles (only the first word should be with a capital first letter), useless mention of months and day of publication (although not consistently), mention of recent or older access to some publications (uninformative), some references incomplete… Moreover, less than 20% of the long list of references appeared within the past 3 years.

R= The format of the references is in accordance with the journal's recommendations and the Mendeley Reference Manager Software by Elsevier was used to insert them, selected the citation style IEEE. probably in the change to the journal's format of the manuscript the changes were made. It was reviewed and modified according to the recommendations. The references as indicated in the modified introduction include the review of the last 10 years, a date in which there is no report of a review with a related topic.

Round 2

Reviewer 1 Report (Previous Reviewer 2)

Comments and Suggestions for Authors

Thank you for the opportunity to review this revised manuscript. The authors have revised the manuscript in accordance with the comments. But still needs major revised.

Can authors precise the content and focu on p38 molecular, regulation and function just in MM or MM therapy? There are so many general information in the manuscript.

Author Response

p38 MAPK molecular targeting for next-generation multiple myeloma therapy

cancers-2759328

Review 1

Thank you for the opportunity to review this revised manuscript. The authors have revised the manuscript in accordance with the comments. But still needs major revised.:

R: Thank you for your comments and review

1. Can authors precise the content and focus on p38 molecular, regulation and function just in MM or MM therapy? There is so many general information in the manuscript.?

R= Thanks for your comments. The general information about p38 on mechanisms of regulation, expression and activation of p38 is generally in the cellular context and in the section on chemical inhibitors the activity of these was mentioned in the context of cancer in general, both parts are the result of the modifications of the original manuscript based on the reviewers' suggestions. In this new version we focused on what was possible and eliminated what was pertinent that did not focus on MM.

Round 3

Reviewer 1 Report (Previous Reviewer 2)

Comments and Suggestions for Authors

The authors have addressed the concerns raised in the previous review, which has improved the quality of the paper.

This manuscript is a resubmission of an earlier submission. The following is a list of the peer review reports and author responses from that submission.

Round 1

Reviewer 1 Report

Comments and Suggestions for Authors

In this manuscript, the author discussed the recent advances of p38 MAPK and the mechanism associated with resistance or progression in multiple myeloma. Moreover, the author summarized the potential significance of clinical application. My comments are as follows.

1.     From line 104 to line 136, the author describes the p38 MAPK target and interactive gene, however, this part is hard to understand. Please re-organize and add more details or mechanisms on how p38 MAPK interacts with the target gene.

2.     The authors are inconsistent in typing. Sometimes mention “p38 MAPK”, however, change to “MAPK p38“ or just “p38”  somewhere. Please make sure it is consistent.

3.     The rationale and logic of this manuscript probably do not well. Such as part ‘2.1’. The author describes the MAPK p38 structure. However, much sentience related to the regulation mechanism and chemical inhibitor was chaos added to confuse this part.

4.     More details related to the chemical or inhibitor are necessary to be added in this manuscript, which could be better for separating one part.

5.     The outlook or the future expectation based on the p38 MAPK research in the clinically significant are necessary in this manuscript.

6.     The structure of this manuscript needs to be approved.

Reviewer 2 Report

Comments and Suggestions for Authors

This review paper summarized the recent research about the p38 MAPK expression, function, and different isoforms regulation, also the possible clinical application of p38 MAPK signaling pathway in multiple myeloma. The manuscript is generally well-written and clear. I believe this paper is publishable. The only thing is the font, the word size and the line space need to be adjusted. It’s not consistent throughout the whole text.

Reviewer 3 Report

Comments and Suggestions for Authors

Although I think this is an interesting topic I have significant concerns regarding plagiarism and hence the overall integrity of the review and I have therefore not completed my review of the manuscript, but I bring this fundamental issue to the attention of the authors and editors. 

I came across this issue because I was intrigued on line 159 with reference to the protein as Q16539-1. I then read the EBI text on function at Q16539

I found striking and repeated similarity in section structure and in exact phrasing between text on lines 104-131 and the text on the EBI website. I then also found similarity elsewhere to text entries on GeneCards relating to the p38 family.

This is not simply a matter of occasional reuse of a phrase but a systematic lifting of structure and content with some paraphrasing. In the institution I work for this would trigger a disciplinary review if submitted for example for a degree.

Specifically see:

Line 104-108

“Some p38 MAPK targets, which are activated by phosphorylation and 104 then phosphorylate other downstream targets, include RPS6KA5/MSK1 105 and RPS6KA4/MSK2, which can directly activate transcription factors such 106 as CREB1, ATF1, RELA/NFKB3, STAT1, and STAT3, as well as histone H3 107 and the nucleoprotein HMGN1[26], [27].”

Compared to Q16539 InterPro

(https://www.ebi.ac.uk/interpro/protein/UniProt/Q16539/)

Some of the targets are downstream kinases which are activated through phosphorylation and further phosphorylate additional targets. RPS6KA5/MSK1 and RPS6KA4/MSK2 can directly phosphorylate and activate transcription factors such as CREB1, ATF1, the NF-kappa-B isoform RELA/NFKB3, STAT1 and STAT3, but can also phosphorylate histone H3 and the nucleosomal protein HMGN1 (PubMed:9687510, PubMed:9792677).

 Line 116-117

“In the cytoplasm, the p38 MAPK pathway is an important regulator of protein turnover.

Compared to Q16539 InterPro

(https://www.ebi.ac.uk/interpro/protein/UniProt/Q16539/)

In the cytoplasm, the p38 MAPK pathway is an important regulator of protein turnover. 

 Line 119-123

“Thus, the detachment of ectodomains from transmembrane proteins is also regulated by 119 p38 MAPK. In response to proinflammatory stimuli, p38 phosphorylates ADAM17, which 120 is a membrane-associated metalloprotein, and this triggers the detachment of ectodo- 121 mains of ligands belonging to the TGF-alpha family, activating the EGFR signaling path- 122 way and cell proliferation [31].”

Compared to Q16539 InterPro

(https://www.ebi.ac.uk/interpro/protein/UniProt/Q16539/)

Ectodomain shedding of transmembrane proteins is regulated by p38 MAPKs as well. In response to inflammatory stimuli, p38 MAPKs phosphorylate the membrane-associated metalloprotease ADAM17. Such phosphorylation is required for ADAM17-mediated ectodomain shedding of TGF-alpha family ligands, which results in the activation of EGFR signaling and cell proliferation.”

Then

Line 124-131

“FGFR1 is capable of regulating rRNA synthesis and cell growth and is also targeted 124 by p38, by which it induces translocation from the cytosol to the nucleus after its activation 125 mediated by p38 MAPK. In general, a wide variety of transcription factors are activated 126 by p38 through phosphorylation in response to different stimuli [32]–[34]. Therefore, p38 127 MAPK has emerged as an important regulator of gene expression by modulating chroma- 128 tin remodeling, activating histone H3, and promoting the inflammatory response by reg- 129 ulating the expression of IL-6, IL-8, and IL-12b through phosphorylation, which increases 130 accessibility to NF-κB binding sites, increasing their recruitment [35].”

Compared to Q16539 InterPro

(https://www.ebi.ac.uk/interpro/protein/UniProt/Q16539/)

Another p38 MAPK substrate is FGFR1. FGFR1 can be translocated from the extracellular space (???) into the cytosol and nucleus of target cells, and regulates processes such as rRNA synthesis and cell growth. FGFR1 translocation requires p38 MAPK activation. 

And 

The p38 MAPKs are emerging as important modulators of gene expression by regulating chromatin modifiers and remodelers. The promoters of several genes involved in the inflammatory response, such as IL6, IL8 and IL12B, display a p38 MAPK-dependent enrichment of histone H3 phosphorylation on 'Ser-10' (H3S10ph) in LPS-stimulated myeloid cells. This phosphorylation enhances the accessibility of the cryptic NF-kappa-B.

Or the sentence from GeneCards

"This phosphorylation enhances the accessibility of the cryptic NF-kappa-B-binding sites marking promoters for increased NF-kappa-B recruitment."

Elsewhere for example the text is in part lifted from GeneCards for MAPK14 

Line 71-75 "p38α, one of the four isoforms, plays an important role in cell signaling 71 cascades triggered by extracellular stimuli such as stress, proinflammatory 72 cytokines, or direct transcriptional activation. p38α plays a central role in 73 signal transduction pathways and inducing the phosphorylation of a wide 74 range of proteins, estimated between 200 and 300 substrates. "

Compare this and other sections to Genecards entry for MAPK14 under the section Function of MAPK14 gene:

"MAPK14 is one of the four p38 MAPKs which play an important role in the cascades of cellular responses evoked by extracellular stimuli such as pro-inflammatory cytokines or physical stress leading to direct activation of transcription factors.
Accordingly, p38 MAPKs phosphorylate a broad range of proteins and it has been estimated that they may have approximately 200 to 300 substrates each."

Comments on the Quality of English Language

Irrelevant